# Anaphylaxis to Sunflower Seed with Tolerance to Sunflower Oil: A Case Report

**DOI:** 10.3390/medicina57070661

**Published:** 2021-06-27

**Authors:** Jin An

**Affiliations:** Department of Pulmonary, Allergy and Critical Care Medicine, Kyung Hee University Hospital at Gangdong, College of Medicine, Kyung Hee University, Seoul 02477, Korea; anjin7487@gmail.com; Tel./Fax: +82-2-440-7188

**Keywords:** anaphylaxis, sunflower seed, sunflower oil, 2S albumins, lipid transfer protein

## Abstract

Sunflower seeds (Helianthus annuus) are an uncommon source of allergy; however, some cases of allergy to sunflower seeds have been reported. Sunflower seed sensitization occurs to storage proteins (2S albumins) and lipid transfer proteins (LTPs). A 46-year-old female presented three allergic reactions within minutes of consuming sunflower seeds. A prick-to-prick test indicated a positive reaction only to sunflower seeds and a negative reaction to other nuts, such as almond, hazelnut, pistachio, cashew, peanut, macadamia, sesame, and walnut. Prick-to-prick and oral provocation tests of sunflower oil were performed, and a negative result was obtained. The patient was prescribed a 0.3 mg epinephrine autoinjector device for emergency intramuscular administration. The patient is currently under avoidance of sunflower seed but eats food cooked in sunflower seed oil. Based on this case, we should recognize that sunflower seeds have the potential to cause severe anaphylaxis, which indicates tolerance to sunflower oil. An accurate and fast diagnosis allows timely recommendation to practice strict avoidance of sunflower seeds, thus reducing the possibility of recurrence of an anaphylactic reaction.

## 1. Introduction

Sunflower seeds (*Helianthus annuus*) are often consumed in the form of seeds, oil, margarine, and bread products, but have rarely been reported to cause anaphylaxis after ingestion [1,2,3,4]. Symptoms of an allergic reaction to sunflower include bronchial asthma, allergic rhinitis, angioedema, acute urticaria, and oral allergy syndrome [5]. The main allergens described in sunflower seed are the 2S albumin protein and a nonspecific lipid transfer protein (LTP), including Hel a 1, Hel a 2, Hel a 3, and Hel a 6 [6,7,8]. Sunflower seed allergen components have been known to contain proteins cross-reactive to mugwort pollen, such as Art v 1 and Art v 3 [9]. The processing of edible oils usually alters the proteins present in sunflower oil, affecting solubility and resulting in a dramatic decrease in total protein content [10]. Here, we report the case of a 46-year-old female who had three anaphylactic events due to sunflower seeds but had tolerance to sunflower oil.

## 2. Case Report

A 46-year-old female visited an allergist to evaluate three events of anaphylaxis. These events had happened within a few minutes of eating sunflower seeds. In the past, the patient had no allergic reactions after ingestion of several nuts, such as peanuts, almonds, and walnuts. She had no history of atopy, thyroid disease, or hepatitis. There was no family history of allergic diseases. She also did not have a pet at home.

The first episode of allergy onset was after ingestion of several dried seeds, including sunflower seeds, 2 years ago. The patient complained of moderate allergic reaction with symptoms of dyspnea and oropharyngeal pruritus within about 5 min of eating the dried seeds. She visited the emergency department and regressed on administration of antihistamine and corticosteroids. The second episode occurred 5 months after the first event. The patient presented a moderate degree of allergic reactions, such as acute urticaria, oropharyngeal pruritus, and dyspnea, after eating only sunflower seeds. Twenty minutes after taking an antihistamine, she recovered from the allergic symptoms. The third episode with severe anaphylaxis occurred 13 months after the second event. The patient complained of abdominal pain, urticaria, and lost consciousness within a few minutes of accidentally consuming a salad and a smoothie that included sunflower seeds at dinner. No cofactors were associated. She was admitted to the emergency department, and her blood pressure (BP), heart rate (HR), and peripheral blood oxygen saturation (SpO2) were 85/60 mmHg, 104 bpm, and 95%, respectively. Her consciousness and vital signs recovered after the intramuscular administration of 0.3 mg epinephrine and intravenous administration of 100 mg hydrocortisone and 4 mg chlorpheniramine maleate. She suspected an allergic reaction to sunflower seeds because three events occurred within a few minutes of ingestion of sunflower seeds.

Laboratory results revealed an increase in the serum levels of total IgE (208 IU/mL), while the other parameters in the laboratory findings were within their normal limits. Skin prick tests with common inhalant allergens, including *Alternaria tenuis*, *Cladosporium herbarum*, *Aspergillus fumigatus*, *Penicillium notatum*, *Ficus benjamina*, *Dermatophagoides pteronyssinus*, *Dermatophagoides farinae*, dog, cat, grass, rye, birch, alder, hazel, ash, beech, mugwort, and latex, showed negative results. A prick-to-prick test was performed with various raw nuts and seeds that had not been roasted. The result revealed a strongly positive reaction to sunflower seeds with the size of the reaction being 7 × 8 (control, 3 × 3); however, the results with other nuts and seeds, including almond, hazelnut, pistachio, cashew, peanut, macadamia, sesame, and walnut, were negative. Fifteen minutes after the prick-to-prick test, the patient complained of itching sensation and dizziness, and hypotension was observed. She recovered after the administration of an antihistamine.

In addition to the sunflower seed allergy assessment, we planned an assessment of her allergic reaction to sunflower oil because the patient was apprehensive about consuming sunflower oil. The result of the prick-to-prick test with fresh sunflower oil was negative. An open oral provocation test with sunflower oil was performed to accurately determine whether the sunflower oil that the patient ingested would elicit an allergic reaction. The oral provocation test with sunflower oil was conducted with a dose equivalent to the dose she normally used in her food (about 5 mL). The patient ate the same amount of sunflower oil three times with an interval of 30 min between each consumption, and she did not complain of any symptoms.

The patient is currently under strict avoidance of sunflower seed but eats food cooked in sunflower seed oil. She was prescribed a 0.3 mg epinephrine autoinjector device for emergency intramuscular administration.

## 3. Discussion

Food allergies have become an important health concern and are associated with a significant negative impact on the quality of life [11]. In the diagnosis of any allergic reaction, a thorough study of the patient’s history is important to identify the causative allergen. However, sunflower seeds are often incorporated in various foods, and the patient may not be aware of it [8]. Fortunately, we suspected the patient’s causative food item relatively easily by studying a detailed history of food intake because she had only eaten sunflower seeds before developing allergic reactions. Additionally, this patient had a strong positive test result in the prick-to-prick test with sunflower seed. Although verification by oral provocation testing remains the gold standard in the diagnosis of food allergy, we did not need to perform the oral challenge test because this patient already experienced allergic reactions after eating sunflower seeds alone. Therefore, the evaluation of food allergy requires thorough history and physical examination to consider a broad differential diagnosis and to ascertain possible trigger foods.

Seeds are now recognized as major food allergens with a potential for anaphylaxis. Sunflower seeds are extensively used in the form of oil, in margarine, and in bread. Although few clinical reports have indicated that sunflower seeds cause anaphylaxis in subjects who are allergic to sunflower seeds, some cases of anaphylaxis due to sunflower seeds have been reported [1,2,12]. The first case of sunflower seed allergy was reported in 1979 [1]. In 1986, Halsey et al. reported two patients allergic to sunflower seeds; their allergies were objectively confirmed with skin prick tests and a radioallergosorbent (RAST) test. Moreover, the authors tested samples of cold-pressed sunflower oil and found traces of protein, but during an open-label sunflower oil provocation, the patients did not complain of any clinical symptoms [3].

The nature of the sunflower allergen has so far been known relatively poorly. The 2S methionine-rich protein (SSA) and LTPs, including Hel a 1, Hel a 2, and Hel a 3, have been described as the main allergens in sunflower seeds [6]. LTPs are recognized as plant food allergens and found in peanuts and certain tree nuts, such as hazelnut, walnut, and almond, and related to cross-reactivity with a variety of vegetables and fruits [7,9]. Some reports suggest possible cross-reactivity between sunflower seeds and other foods that contain LTPs related to mugwort pollen [8]. In this study, the patient did not demonstrate cross-reactivity with other seeds and pollens containing LTPs (hazel and mugwort) considering the negative results of skin prick with common inhalant allergens and prick-to-prick tests with various seeds. This may suggest that cross-reactivity with LTPs from botanically unrelated seeds is not always the case [13]. It is important to recognize that cross-reactivity with LTPs may or may not reflect clinical allergic symptoms, emphasizing the importance of verification of clinical allergy by prick-to-prick and oral provocation tests.

An individualized oral provocation test with sunflower oil was conducted on this patient, taking into account the anticipated clinical reaction and the causative quantity. It was concluded that this patient who experienced anaphylaxis to sunflower seeds presented a negative result of the oral provocation test for sunflower seed oil. Sunflower oil may rarely cause allergic reactions because it is more commonly tolerated due to extremely low protein content after processing [14]. The allergenicity of the proteins present in the oil is gradually modified at the refining stage. Most patients with allergy to sunflower seeds tolerate the small amount of allergenic protein contained in sunflower seed oil. Although anaphylactic reactions to sunflower seed oil have been known to occur rarely, some case studies reported patients with allergic reaction to sunflower seed oil [14]. The patients experienced three episodes of anaphylaxis starting 1 h after eating a meal containing sunflower oil and margarine. A food challenge was performed with 5 mL sunflower oil, and cough, impaired respiration, and angioedema developed. Another study also reported allergic reaction to sunflower oil, and the authors suggested that low levels of residual protein in sunflower oil does not rule out the possibility of allergic reactions in highly sensitized subjects [15]. The occurrence of allergic reactions to sunflower oil may be explained by LTP sensitization [14,15]. All LTPs are potent pan-allergens, and their biological activity is favored by their resistance to heat and gastric reactions [7]. Therefore, sensitization to these may be the cause of anaphylaxis on ingestion of sunflower seed oil cooked at high temperatures.

This case study has some limitations. The patient’s diagnostic evaluations were not expanded to specific IgE level and allergen component testing with the use of ImmunoCAP. In addition, we did not confirm protein profiles of sunflower seed extracts by sodium dodecyl sulfate polyacrylamide gel electrophoresis (SDS-PAGE).

## 4. Conclusions

We present a case of anaphylaxis due to sunflower seed ingestion with tolerance to sunflower oil. This may occur because the processing of sunflower oil induces protein content reduction. Sunflower seed allergy should be suspected by careful observation of the history of food intake and confirmed by prick-to-prick and oral provocation tests. An accurate and fast diagnosis allows timely recommendation to practice strict avoidance of sunflower seeds, thus reducing the possibility of recurrence of an anaphylactic reaction.

## Data Availability

The data reported in this paper are available from the medical history of the patient.

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
