# Peer review of "Anaphylaxis to Sunflower Seed with Tolerance to Sunflower Oil: A Case Report"

_medicina, 2021, doi:10.3390/medicina57070661_

Round 1
Reviewer 1 Report
Interesting article worth publishing.
However, in the case report and the currently performed tests - in fact basic in allergology - it cannot be concluded which molecule the patient is allergic to. There are at least 4 - immunogenic sunflower proteins, it is necessary to describe them in the introduction.
In the discussion, one should rather exclude an allergy to LPT - the patient does not react to other food products - where LPT is common, nor does he react to nuts.
On the other hand, allergy to LPT - often requires a cofactor, which is usually exercise, so a negative challenge with oil, where traces of sunflower proteins are present, could be a false negative.
Have other causes of anaphylaxis ruled out - salad? What ingredients-? what is your usual blood pressure? The criterion for anaphylaxis was a reduction of 20 mmHg or 20% from baseline blood pressure
Work limit - determination of the tryptase level - were there any trials of 3 anaphylactic reactions, 1 time after the skin test, was it done for the principle of prick by prick with the ingredients of the salad?
Was a sunflower seed test performed in a healthy volunteer? to rule out non-specific skin reactivity
It is necessary to indicate according to one of the available classifications of the degree of anaphylaxis, or at least to exclude that it was not the 1st degree of anaphylaxis (mild).
Author Response
Interesting article worth publishing.
However, in the case report and the currently performed tests - in fact basic in allergology - it cannot be concluded which molecule the patient is allergic to. There are at least 4 - immunogenic sunflower proteins, it is necessary to describe them in the introduction.
(Response) Based on the reviewer’s comment, we have been added the content of sunflower proteins in the introduction as follows:
(Introduction) The main allergens described in sunflower seed are 2S albumin protein and a nonspecific lipid transfer protein (LTP), including Hel a 1, Hel a 2, Hel a 3, and Hel a 6. Sunflower seed allergen components have been known to contain proteins cross-reactive with mugwort pollen, such as Art v 1 and Art v 3.
We have added the reference.
Wawrzenczyk, A.; Zbikowska-Gotz, M.; Wawrzenczyk, A.; Bartuzi, Z. Sensitisation to lipid transfer proteins in pollen - allergic adults with food allergy. Postepy Dermatol Alergol 2020, 37, 508-512.
In the discussion, one should rather exclude an allergy to LPT - the patient does not react to other food products - where LPT is common, nor does he react to nuts.
(Response) We appreciate and agree with this comment. I rewrote the discussion section related to LTPs as follows:
(Discussion) LTPs are recognized as plant food allergens and found in peanuts and certain tree nuts, such as hazelnut, walnut, and almond, and related to the cross-reactivity with a variety of vegetables and fruits. Some reports suggested possible cross-reactivity between sunflower seeds and other foods that contain LTPs related to mugwort pollen. In this study, the patient did not demonstrate cross-reactivity to other seeds and pollens containing LTPs (hazel and mugwort) through the results of skin prick with common inhalant allergens and prick-to-prick tests with various seeds. This may suggest that cross-reactivity to LTPs from botanically unrelated seeds is not always the case. It is important to recognize that cross-reactivity to LTPs may or may not reflect clinical allergic symptoms, emphasizing the importance of verification of clinical allergy by prick-to-prick and oral provocation test.
On the other hand, allergy to LPT - often requires a cofactor, which is usually exercise, so a negative challenge with oil, where traces of sunflower proteins are present, could be a false negative.
(Response) We considered the possibility of FDEIA with sunflower seed oil; however, we did not perform the oral food provocation test with sunflower oil followed by physical exercise because (1) there were not cofactors associated with anaphylaxis of sunflower seed and (2) the patient refused to conduct exercising test due to her weakness. Although an oral sunflower oil provocation test without the addition of cofactors, such as NSAIDs and exercise was performed, we could suspect clinically that this patient did not have a sunflower oil allergy because she has eaten sunflower seed oil and did not complain of any allergic symptoms for a long time. Regarding the reviewer’s comment, we have added the sentence related to cofactors in case report as follows:
(Case report) Not cofactors were associated.
Have other causes of anaphylaxis ruled out - salad? What ingredients-? what is your usual blood pressure? The criterion for anaphylaxis was a reduction of 20 mmHg or 20% from baseline blood pressure
(Response) We could rule out other causes of anaphylaxis with her second episode, which is allergic reactions after eating only sunflower seeds. This patient’s usual blood pressure was 120-130/70-80 mmHg. We could confirm anaphylaxis with a reduction of BP.
Work limit - determination of the tryptase level - were there any trials of 3 anaphylactic reactions, 1 time after the skin test, was it done for the principle of prick by prick with the ingredients of the salad?
(Response) Unfortunately, the tryptase level was not checked at first and third events visited the emergency department. However, we could clinically diagnose anaphylaxis with symptoms and signs.
We considered and recommended the prick by prick test with the ingredients of the salad, but the patient did not remember clearly except for peanut, almond, and lettuce. She ate lettuce afterward but had no allergic reactions.
Was a sunflower seed test performed in a healthy volunteer? to rule out non-specific skin reactivity
(Response) Unfortunately, we did not perform skin prick and prick by prick tests in healthy subjects. However, we performed the skin tests that included controls with saline (negative control) and histamine (positive control). Especially, we could check non-specific skin reactivity with the result of negative control in each skin test.
It is necessary to indicate according to one of the available classifications of the degree of anaphylaxis, or at least to exclude that it was not the 1st degree of anaphylaxis (mild).
(Response) We would like to thank the reviewer for this comment. The sentences have been added according to the reviewer’s comment as follows:
(Case report) The patient complained of moderate allergic reaction with symptoms of dyspnea and oropharyngeal pruritus within about five minutes of eating the dried seeds.
(Case report) The second episode occurred five months after the first event. The patient presented moderate degree of allergic reactions, such as acute urticaria, oropharyngeal pruritus, and dyspnea, after eating only sunflower seeds.
(Case report) The third episode with severe anaphylaxis occurred 13 months after the second event. The patient complained of abdominal pain, urticaria, and lost consciousness within a few minutes of consuming a salad and a smoothie that included sunflower seeds accidentally, at dinner.
Reviewer 2 Report
This is an interesting and well written case study. I have just one minor recommendation for the author to consider. That is to list in the Abstract all nuts tested, not just some with etc.
Author Response
This is an interesting and well written case study. I have just one minor recommendation for the author to consider. That is to list in the Abstract all nuts tested, not just some with etc.
(Response) Based on the reviewer’s suggestion, I have changed the sentence of nuts test in the abstract as follows:
(Abstract) A prick-to-prick test was indicated a positive reaction only to sunflower seeds and was negative to other nuts, such as almond, hazelnut, pistachio, cashew, peanut, macadamia, sesame, and walnut.